# Effects of a Paediatric Antimicrobial Stewardship Program on Antimicrobial Use and Quality of Prescriptions in Patients with Appendix-Related Intraabdominal Infections

**DOI:** 10.3390/antibiotics10010005

**Published:** 2020-12-23

**Authors:** Sílvia Simó, Eneritz Velasco-Arnaiz, María Ríos-Barnés, María Goretti López-Ramos, Manuel Monsonís, Mireia Urrea-Ayala, Iolanda Jordan, Ricard Casadevall-Llandrich, Daniel Ormazábal-Kirchner, Daniel Cuadras-Pallejà, Xavier Tarrado, Jordi Prat, Emília Sánchez, Antoni Noguera-Julian, Clàudia Fortuny

**Affiliations:** 1Infectious Diseases and Systemic Inflammatory Response in Paediatrics, Infectious Diseases Unit, Department of Paediatrics, Sant Joan de Déu Hospital Research Foundation, 08950 Barcelona, Spain; ssimo@fsjd.org (S.S.); evelasco@sjdhospitalbarcelona.org (E.V.-A.); mrios@sjdhospitalbarcelona.org (M.R.-B.); cfortuny@sjdhospitalbarcelona.org (C.F.); 2Centre for Biomedical Network Research on Epidemiology and Public Health (CIBERESP), 28029 Madrid, Spain; ijordan@sjdhospitalbarcelona.org; 3Pharmacy Department, Sant Joan de Déu Hospital, 08950 Barcelona, Spain; mlopezr@sjdhospitalbarcelona.org; 4Clinical Microbiology Department, Sant Joan de Déu Hospital, 08950 Barcelona, Spain; mmonsonis@sjdhospitalbarcelona.org; 5Patient Safety Area—Infection Control Unit, Sant Joan de Déu Hospital, 08950 Barcelona, Spain; murrea@sjdhospitalbarcelona.org; 6Pediatric Intensive Care Unit, Sant Joan de Déu Hospital, 08950 Barcelona, Spain; 7Department of Paediatrics, University of Barcelona, 08007 Barcelona, Spain; 8Management Department, Sant Joan de Déu Hospital, 08950 Barcelona, Spain; ricard@sjdhospitalbarcelona.org; 9Computing Department, Sant Joan de Déu Hospital, 08950 Barcelona, Spain; dormazabal@sjdhospitalbarcelona.org; 10Statistics Department, Sant Joan de Déu Research Foundation, 08950 Barcelona, Spain; dcuadras@fsjd.org; 11Paediatric Surgery Department, Sant Joan de Déu Hospital, 08950 Barcelona, Spain; xtarrado@sjdhospitalbarcelona.org (X.T.); joprat@sjdhospitalbarcelona.org (J.P.); 12Blanquerna School of Health Science, Ramon Llull University, 08022 Barcelona, Spain; emiliasr@blanquerna.url.edu; 13Translational Research Network in Paediatric Infectious Diseases (RITIP), 28009 Madrid, Spain

**Keywords:** antimicrobial stewardship, appendicitis, children, days of therapy, intra-abdominal infection, length of stay, paediatric

## Abstract

The effectiveness of antimicrobial stewardship programs (ASP) in reducing antimicrobial use (AU) in children has been proved. Many interventions have been described suitable for different institution sizes, priorities, and patients, with surgical wards being one of the areas that may benefit the most. We aimed to describe the results on AU and length of stay (LOS) in a pre-post study during the three years before (2014–2016) and the three years after (2017–2019) implementation of an ASP based on postprescription review with feedback in children and adolescents admitted for appendix-related intraabdominal infections (AR-IAI) in a European Referral Paediatric University Hospital. In the postintervention period, the quality of prescriptions (QP) was also evaluated. Overall, 2021 AR-IAIs admissions were included. Global AU, measured both as days of therapy/100 patient days (DOT/100PD) and length of therapy (LOT), and global LOS remained unchanged in the postintervention period. Phlegmonous appendicitis LOS (*p* = 0.003) and LOT (*p* < 0.001) significantly decreased, but not those of other AR-IAI diagnoses. The use of piperacillin–tazobactam decreased by 96% (*p* = 0.044), with no rebound in the use of other Gram-negative broad-spectrum antimicrobials. A quasisignificant (*p* = 0.052) increase in QP was observed upon ASP implementation. Readmission and case fatality rates remained stable. ASP interventions were safe, and they reduced LOS and LOT of phlegmonous appendicitis and the use of selected broad-spectrum antimicrobials, while increasing QP in children with AR-IAI.

## 1. Introduction

Antimicrobial stewardship programs (ASP) are one of the interventions that have been proved to reduce antimicrobial use (AU) in children [1,2,3,4,5,6,7]. Numerous different ASP interventions exist, designed for different health centres, settings, clinical syndromes, and patients. However, data on the most effective stewardship strategies are limited for the paediatric population, while standardised and validated clinical endpoints are still under discussion [1,2,8,9]. Surgical wards have been described as one of the areas that may benefit the most from ASP interventions [10,11,12]. Improving the quality of surgical prescriptions, both in prophylactic and therapeutic protocols, may have a relevant impact on patient outcomes, antimicrobial resistance rates, and hospital expenses [13].

Recent publications demonstrate that shorter courses of antimicrobial therapy are appropriate for complicated intraabdominal infections (IAIs) and do not lead to an increase in the number of readmissions, nor in morbidity and mortality rates, especially following timely surgical intervention and source control [14,15]. The Infectious Diseases Society of America guidelines recommend treatment for 4 to 7 days in the setting of adequate source control in IAIs [16]. Several randomised clinical trials have shown that 3 to 4 days of antibiotics in a patient with clinical improvement is as effective and safe as longer courses of therapy, again provided that source control is achieved [17,18,19]. In spite of the available scientific evidence, broad-spectrum antibiotic overuse and excessive duration of antibiotic therapy persist in surgical wards [14,20].

The appendix is the most common source of infection in community-acquired IAIs, especially among children and adolescents [21]. A single prophylactic antimicrobial dose is recommended for noncomplicated (nonperforated) phlegmonous appendicitis before intervention [22,23]. However, there is wide variation among centre guidelines and protocols with respect to antibiotic regimens recommended in complicated paediatric appendicitis (defined as gangrenous perforated appendicitis, with or without local or generalised peritonitis, abscess, or appendiceal mass), and the optimal antibiotic regimen and duration of therapy are still unclear [10]. The most commonly identified bacteria in perforated appendicitis in children are anaerobes and *Escherichia coli,* followed by *Streptococcus anginosus* group, *Klebsiella pneumoniae*, and *Pseudomonas aeruginosa*, with extended-spectrum β-lactamase (ESBL) and other multidrug-resistant strains rates depending on local epidemiology [24,25].

We aimed to describe and to evaluate the results on AU, length of stay (LOS) and quality of prescriptions (QP) of the first 3 years of an ASP intervention directed to children admitted for appendix-related intraabdominal infections (AR-IAI) in a European referral paediatric university hospital.

## 2. Results

Overall, 2021 admissions between January 2014 and December 2019 (919 in the preintervention period and 1102 in the postintervention period) for AR-IAIs were included (44.1% phlegmonous appendicitis, 28.9% gangrenous appendicitis, 24.4% appendicular peritonitis, 1.5% appendicular abscesses, and 1.1% appendicular masses). Patients median (IQR) age at diagnosis was 10.0 (7.4–12.8) years in the preintervention period and 10.4 (7.9–13.0) years in the postintervention period (*p* = 0.241; Appendix A). An increase in the total number of admissions due to AR-IAI in the post-intervention period was observed, together with a significant change in the distribution of the different diagnoses between the two periods (*p* < 0.001, chi-square test; rates of gangrenous appendicitis and appendicular masses increased, while rates of phlegmonous appendicitis and appendix-related peritonitis decreased; Appendix A). Laparoscopy was the most common surgical approach (*p* = 0.255) both in the preintervention period (93.0%) and in the postintervention period (93.3%). No patient died during the study period.

### 2.1. Days of Therapy (DOTs), Length of Therapy (LOT), and Length of Stay (LOS)

The main results are summarised in Table 1. In an interrupted time series analysis, no significant changes between periods were observed in global AU, expressed both in days of therapy/100 patient days (DOT/100PD; *p* = 0.113) and LOT (*p* = 0.298, Appendix A), or in global LOS (*p* = 0.314, Appendix A). A significant reduction in LOS (*p* = 0.003) and LOT (*p* < 0.001, Appendix A) was observed in phlegmonous appendicitis, but not in the rest of the diagnoses. A significant decrease in the monthly use of piperacillin–tazobactam [median (IQR) values: 44.2 DOT/100PD (39.1–49.8 DOT/100PD) in 2014–2016 vs. 1.9 DOT/100PD (0.0–9.3 DOT/100PD) in 2017–2019; *p* = 0.044] occurred, in favour of cefoxitin, ceftriaxone, and metronidazole use, although changes in the use of the latter did not reach statistical significance (Appendix A). LOT and LOS for the different AR-IAIs diagnoses are shown in Appendix A, respectively.

### 2.2. Quality of Prescriptions

Overall, 715 antimicrobial prescriptions from 573 different admissions were evaluated during 2017–2019, representing 52% (573/1102) of all AR-IAI admissions in this period. The QP evaluation was performed by the ASP team at a median (IQR) time of 36.3 (33.9–80.9) hours after the initial prescription. Globally, 80.1% (*n* = 593) of the prescriptions were considered optimal. The diagnosis with the worst QP rates was phlegmonous appendicitis, with only 54.0% of optimal prescriptions (*p* < 0.001, chi-square test; Appendix A). The most frequent reasons for a prescription to be considered nonoptimal (*n* = 122) were: excessive duration of treatment (*n* = 78, 63.9%), noncompliance with the local protocol (*n* = 44, 36.1%), inadequate antimicrobial spectrum (*n* = 37, 30.3%), wrong dosing (*n* = 22, 18.0%) and absence or delay in sequential treatment (*n* = 17, 13.9%). A trend towards an improvement in QP rates was observed during the postintervention period (*p* = 0.052, Figure 1).

### 2.3. Other Outcomes

The ASP implementation had no impact in readmission rates (RR) or case fatality rates (FR) over time (Table 1).

## 3. Discussion

Surgery departments have been identified as those that may benefit the most from ASP, most often because of prescriptions including unnecessary prolonged antibiotic regimens and unnecessary use of wide-spectrum antibiotics, both in adult [11] and paediatric patients [10]. To date, the literature on the implementation of ASP in children and adolescents affected with surgical conditions is still scarce and uses different strategies and metrics that preclude direct comparisons between studies [9,11,21]. Our study analyses the impact of the first 3 years of a paediatric ASP based on postprescription review with feedback (PPRF) on AU, LOS, and QP in children and adolescents with AR-IAIs. The intervention significantly reduced LOS and LOT in phlegmonous appendicitis, but not in the rest of diagnoses. Global AU, expressed in DOT/100PD, remained unchanged upon ASP implementation. Global LOT and LOS decreased in the postintervention period, albeit these changes were not statistically significant. Piperacillin–tazobactam use dramatically decreased without any rebound in the use of carbapenems or other wide-spectrum antibacterials. The stability in RR and the lack of deaths during the whole study period indicate patient safety of our ASP. These results are consistent with previous studies in adult patients [10,14,15] and confirm the safety and efficacy of ASP also in the most prevalent surgical paediatric patients, those affected with AR-IAIs. It has been previously demonstrated that ASP based even solely on passive institutional guideline implementation can improve AU in IAIs. Popovski et al. implemented a guideline with the objective of limiting antipseudomonal therapy in adults with community-acquired IAIs. They reported reductions in DOT of ciprofloxacin and piperacillin/tazobactam together with increases in ceftriaxone use, without significant impacts on clinical outcomes, mortality, or hospital readmission rates [26]. Dubrovskaya et al. also reported reductions in the use of broad-spectrum antimicrobial agents in adults along with increased use of cefoxitin in a similar study; mortality and LOS remained similar between groups [27]. Finally, Skarda et al. implemented a guideline focused on optimizing DOT based on clinical response and earlier oral step-down therapy in paediatric patients with acute appendicitis [28] and reported that the number of postoperative antibiotic doses decreased, more patients transitioned early to oral therapy, and LOS and global hospital cost decreased without differences in hospital readmission, consultation with the emergency department, or reoperation rates. We designed an ASP based on PPRF, which has a stronger impact on decreasing AU compared with passive interventions or preprescription authorisation, as per the evidence available in adults [29]. Our program included 2 core strategies: interaction and feedback with the prescriber and preauthorisation requirements only for specific agents. Restrictive approaches are also at risk of creating a conflict in antimicrobial management in some wards, which is avoidable with educational and persuasive action [30].

We did not observe changes in global AU in the postintervention period. AU measured in DOT/100PD slightly increased and AU measured in LOT and LOS showed a decreasing trend, albeit none of these changes were statistically significant. We could not demonstrate that our ASP reduced these parameters; nevertheless, the reduction in LOT and LOS over time may still be relevant for patients’ quality of life and hospital management issues [13,31]. These findings are at least partially attributable to an increase in gangrenous appendicitis admission rates (that require a 3-day to 5-day course of antibiotics) together with a decrease in phlegmonous appendicitis admission rates (maximum 1-day course of antibiotics), but also to a change in the peritonitis treatment protocol from 2017 onwards. The new regimen for generalised peritonitis included two narrower spectrum drugs (ceftriaxone and metronidazole, the use of which increased upon ASP implementation) instead of a single antimicrobial agent with a broader antimicrobial spectrum (piperacillin–tazobactam), which resulted in a twofold increase in DOT for this diagnosis. We want to stress that the use of piperacillin–tazobactam in the postintervention period dramatically decreased, without a parallel increase in the use of other antipseudomonal drugs, such as meropenem. The latter may lead to a decrease in antimicrobial resistance rates in AR-IAI patients in the long term, one of the ultimate goals of ASP [1,2,3,9]. Actually, reductions in the prevalence rates of infections by *Clostridium difficile* and multi-drug resistant *Pseudomonas* spp. have been reported in the long term [32,33]. An intrinsic limitation of both DOT and LOT is that differences between one or more doses given in the same day are not reflected [34]. As an example, DOT and LOT account 1 both for a patient receiving a single dose of antimicrobial at 8 a.m. the day of discharge and another receiving two different antimicrobials every 6 h for a full day (8 doses). Similarly, these indicators did not allow us to accurately assess whether phlegmonous appendicitis prescriptions were shortened from 24 h of treatment duration to one single antimicrobial dose as recommended in guidelines [22,23]. More than one metric is always advisable when measuring AU to partially overcome these limitations, since the use of specific drugs or group of drugs separately is often needed to properly identify changes in the spectrum of prescribed antimicrobials [34].

The CODA Collaborative Study has recently demonstrated that the conservative management with antibiotics of adults affected with appendicitis is not inferior to surgical appendectomy [35]. A previous meta-analysis had assessed this therapeutic strategy in children and concluded that, while the conservative management was effective in the short-term, it associated a higher failure rate with conservative management later on and recommended surgery as the treatment of choice for uncomplicated appendicitis in children [36]. To date, the conservative management of AR-IAI in our centre is uncommon and usually reserved for older children with appendicular abscesses or masses that receive intravenous antibiotics initially and undergo second-step surgery some weeks later. In consequence, we were not able to assess the impact of the conservative approach in AU and LOS in our study.

QP improved throughout the postintervention period with borderline statistical significance (*p* = 0.052). Our ASP was able to evaluate over 50% of all prescriptions within the recommended timeframe of 48–72 h after the initial prescription [3,37]. As previously described [38,39], the most frequent reason for a prescription being considered nonoptimal was excessive duration of treatment. In our case, this was especially striking in phlegmonous appendicitis. Even though both LOT and LOS significantly decreased upon ASP implementation for this condition, it was often treated with more than a single perioperative antimicrobial dose in the POSTintervention period as recommended in the guidelines [22]. The qualitative improvement in antimicrobial prescriptions regardless of the decrease in AU is one of the main objectives of ASP [3]. Previous studies have shown long-term improvements in QP after ASP implementation [40,41], highlighting the importance of institutional support and continuous ASP activities in health care centres. We would also like to emphasise the importance of the multidisciplinary collaboration with the surgeons and other specialists [1,3], as well as the critical support roles played by the computer, statistics, and hospital management teams in our centre, which should never be overlooked.

Our study is limited by the observational design, the short follow-up time, and the fact that we focused on AR-IAIs, a prevalent but also specific surgical diagnosis. Additionally, no data on antimicrobials prescribed at discharge were recorded. We were unable to assess neither the financial results of our intervention nor potential changes in antimicrobial resistance rates for this specific group of patients, both of which are complementary measures of many ASPs. Finally, we did not measure prescriber satisfaction or the level of acceptance of ASP recommendations, despite a subjective perception that most surgeons were satisfied and showed increasing willingness to collaborate with the Paediatric Infectious Diseases Unit and the ASP team. The acceptance of prescribers is essential to achieving better results [3,30].

## 4. Materials and Methods

### 4.1. Study Design

This was a pre-post study comparing systemic (intravenous, intramuscular, and enteral) AU and LOS in paediatric inpatients admitted for AR-IAIs after the implementation of an ASP based on PPRF; the preintervention period (2014–2016) was compared with the postintervention period (2017–2019). In the postintervention period, the QP was also assessed. RR and FR were evaluated as quality and safety measures.

### 4.2. Setting and Patients

The study was conducted in the surgical ward of Hospital Sant Joan de Déu, a 268-bed referral tertiary care university hospital for patients below 18 years of age in Barcelona, Spain, with a full range of paediatric medical and surgical subspecialties, a 28-bed paediatric intensive care unit and a 38-bed neonatal intensive care unit. In our centre, appendicitis is diagnosed on the basis of consistent history and physical examination, compatible selected laboratory studies and, in most cases, abdominal ultrasound evaluation [42]. Laparoscopic appendectomy is the treatment of choice over open appendectomy in most AR-IAI patients; conservative management of appendicitis is rare in our institution. All patients admitted because of acute phlegmonous or gangrenous appendicitis, appendicular peritonitis, or postappendicitis abdominal abscess or appendiceal mass, and who received at least one dose of a systemic antimicrobial during admission, were eligible. Negative appendectomies and AR-IAI in immunosuppressed patients were excluded.

### 4.3. Intervention: PROA-SJD

Our ASP (Programa de Optimización del uso de Antimicrobianos Sant Joan de Déu, PROA-SJD) was implemented in January 2017 [7]. The ASP core team was composed of a full-time paediatric infectious diseases specialist, and part-time physicians including a paediatric intensive care specialist, a clinical pharmacist, a microbiologist and an infection control and hospital epidemiology physician. Support was received from computer, statistics, and management hospital teams. Information Technology tools for daily work and for subsequent data analysis were established.

The local guidelines for AR-IAI were discussed together with the surgical team and updated on January 2017, the first month of the ASP implementation. The recommended antimicrobial regimens in local protocols for AR-IAIs in the preintervention period (2014 to 2016) and upon ASP implementation in the postintervention period (2017 to 2019) are summarised in Table 2. The latter were based on the Guidelines by the Surgical Infection Society and the Infectious Diseases Society of America [16].

The core strategy of the ASP was PPRF (summarised in Figure 2). Based on evaluation of the main aspects of diagnosis and treatment, antimicrobial prescriptions were considered “optimal” or “nonoptimal” [7,44,45,46]. For a prescription to be considered “optimal”, all the following criteria had to be met: (1) the administration of the antimicrobial was appropriate considering the diagnosis, antimicrobial spectrum, and our own protocols (Table 2), was adapted to local epidemiology, and also accounted for patient allergies and comorbidities; (2) the drug was given via the right route, and at the right dose and with the right schedule; and (3) the expected and/or actual duration of the antimicrobial treatment was appropriate [7]. For “nonoptimal” prescriptions, feedback with recommendation to discontinue or to modify therapy was provided daily in the patient’s electronic clinical chart and also face-to-face or by phone during morning rounds with the surgical team. Acceptance of recommendations was at the prescribers’ discretion. No preprescription authorisation strategies were implemented, but prescription filters for selected antimicrobial agents (meropenem, linezolid, teicoplanin, colistin, liposomal amphotericin B, itraconazole, voriconazole, posaconazole, micafungin, gancyclovir, cidofovir, valganciclovir, and foscarnet) were incorporated in the e-prescription system, forcing prescribers to indicate the reason for selecting one of these drugs before validation of the prescription. In addition, pre-set protocols with automatic calculation of dosing according to patient weight for the different AR-IAIs were included in the e-prescription program. Finally, the ASP team also attended quarterly team meetings to discuss specific aspects of AU and to give direct feedback to the paediatric surgery team.

### 4.4. Definitions

IAIs were classified as phlegmonous appendicitis, gangrenous appendicitis, appendicular peritonitis, appendicular abscess, and appendicular mass as per the surgeons operative and clinical reports.

#### 4.4.1. Days of Therapy (DOTs), Length of Therapy (LOT), and Length of Stay (LOS)

AU data were expressed as DOT/100 PD. DOTs were defined as the number of days that a patient received each antibiotic, regardless of the dose or number of doses. When a patient received more than one antibiotic, more than one DOT was counted [47]. Antibacterial administration data were extracted from the e-prescription program during the admission. Oral antibiotics prescribed at discharge used different e-prescription software and were not collected. DOTs were totalled for each month and then standardised to 100 PD using total PD for AR-IAI admissions in a given month. Person time was calculated in PD by subtracting date of discharge from date of admission. An individual patient counted 1 PD on each calendar day; between-unit transfers did not result in double counting [48].

Length of therapy (LOT) was defined as the number of consecutive days that a patient received systemic antimicrobial agents, irrespective of the number of antibiotics or doses [47]. Mean monthly LOT was calculated by dividing total LOT by the number of admissions due to AR-IAIs in a given month.

Length of stay (LOS) was defined as the length of an inpatient episode of care, calculated from the day of admission to the day of discharge, and based on the number of nights spent in hospital. LOS in patients admitted and discharged on the same calendar day was 1 day [47]. Again, mean monthly LOS was calculated by dividing total LOS by the number of admissions due to AR-IAIs in a given month.

#### 4.4.2. Quality of Prescriptions

During the postintervention period (2017–2019), standardised evaluations of the quality of antimicrobial prescriptions were made as detailed above for patients with AR-IAIs admitted to the surgical ward at 8 a.m. every working day.

#### 4.4.3. Other Outcomes

RR were calculated as the percentage of nonscheduled surgery ward readmissions related to AR-IAI in the 90 days following discharge, and FR were calculated as the percentage of all-cause deaths in the 365 days following discharge; both rates were also collected.

### 4.5. Statistical Methods

Statistical analyses were carried out using SPSS v21.0 software (IBM Corporation, Armonk, NY, USA) and R software (R Development Core Team 2013. R: A language and environment for statistical computing. R Foundation for Statistical Computing, Vienna, Austria. http://www.R-project.org). Categorical variables were reported as proportions with 95% confidence intervals (95% CI), and continuous variables as mean/medians with interquartile ranges (IQRs). Changes in AR-IAI admission rates between the preintervention and the postintervention periods and reasons for nonoptimal QP between the different diagnoses were compared with the chi-square test. Changes in AU (calculated both as DOT/100PD and as LOT), LOS, RR, and FR were assessed using interrupted time series analysis (step change model), showing level changes and slope changes over time. For these analyses, all data were totalled for each month; yearly results (median [IQR] of monthly results) are shown in tables for better data visualisation. Statistical significance was defined as a *p*-value < 0.05.

### 4.6. Ethics Statements

This study was approved by Sant Joan de Déu Research Foundation Ethics Committee [PIC 32-20]. A waiver of the individual’s informed consent was granted. The research was conducted in accordance with the Declaration of Helsinki and national and institutional standards.

## 5. Conclusions

In conclusion, ASP safely reduced LOT and LOS in phlegmonous appendicitis and the global use of piperacillin–tazobactam, while improving antimicrobial QP in children admitted due to AR-IAIs. Our results are consistent with those of previous studies on ASP in the paediatric inpatient and highlight the importance of continuous support of these interventions over time in healthcare centres. Further studies are needed to identify the best combination of AU indicators for measuring these interventions.

## Figures and Tables

**Figure 1 antibiotics-10-00005-f001:**
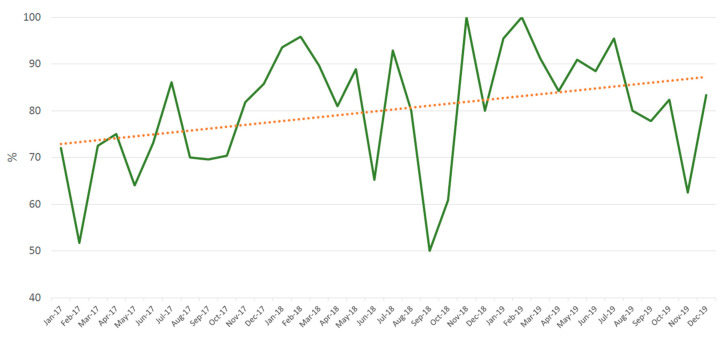
Percentage of optimal prescriptions during the postintervention period (2017–2019) in patients admitted due to appendix-related intraabdominal infections (AR-IAIs). Overall, 715 prescriptions were evaluated and a trend towards an improvement was observed (*p* = 0.052).

**Figure 2 antibiotics-10-00005-f002:**
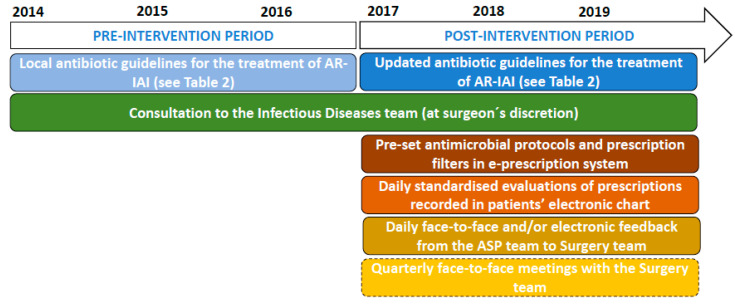
Newly implemented interventions as part of the antimicrobial stewardship program (ASP) to optimise the antibiotic management of children and adolescents admitted due to appendicular-related intraabdominal infections (AR-IAI).

**Table 1 antibiotics-10-00005-t001:** Antimicrobial use (AU), both as days of therapy/100 patient days and as length of therapy (in days), and length of stay (in days) expressed as median (IQR) of monthly results; readmission rates and case fatality rates expressed as percentages (CI95%), totalled for each year for better data visualisation in appendix-related intraabdominal infections (AR-IAI) during the study period. Changes were assessed using interrupted time series analysis (step change model).

	Preintervention Period			Postintervention Period			
Indicator	2014	2015	2016	2017	2018	2019	*p*
Days of therapy/100 patient days	128.1 (118.4–143.6)	122.6 (109.7–132.8)	114.0 (110.6–126.8)	128.8 (122.3–157.2)	132.8 (117.2–140.6)	144.0 (133.8–158.9)	0.113
Length of therapy/AR-IAI (days)	4.7 (4.5–5.2)	4.7 (4.4–5.4)	5.2 (4.8–5.6)	4.5 (4.0–4.6)	4.4 (4.0–5.2)	4.1 (3.9–4.6)	0.298
Length of stay/AR-IAI (days)	4.2 (4.0–4.7)	4.5 (3.9–5.0)	4.6 (3.8–5.1)	4.0 (3.7–4.5)	4.1 (3.7–4.9)	3.6 (3.3–3.9)	0.314
Readmission rates	2.6 (2.0–3.1)	3.2 (2.7–3.9)	5.9 (5.5–6.7)	4.7 (4.1–5.1)	3.5 (2.9–4.0)	5.8 (5.3–6.3)	0.513
Case fatality rates	0.0	0.0	0.0	0.0	0.0	0.0	1.0

**Table 2 antibiotics-10-00005-t002:** Recommended first-line and preferred alternative antimicrobial regimens in appendix-related intraabdominal infections (AR-IAI) during the study periods.

AR-IAI (Recommended Duration of Treatment)	Preintervention Period (2014–2016)	Postintervention Period (2017–2019)
Phlegmonous appendicitis (1 to 3 doses) *	Cefoxitin	Cefoxitin
	Alternative:	Alternative:
	Gentamicin + metronidazole	Gentamicin + metronidazole
Gangrenous appendicitis with or without local peritonitis (3 to 5 days)	Cefoxitin	Cefoxitin
	Alternative:	Alternative:
	Gentamicin + metronidazole	Gentamicin + metronidazole
Appendicular peritonitis (7 to 10 days)	Piperacillin–tazobactam	Ceftriaxone plus metronidazole
	Alternative:	Alternative:
	Gentamicin + metronidazole	Gentamicin + metronidazole or piperacillin–tazobactam
Appendicular abscess or mass (7 to 10 days) **^&^**	Amoxicillin–clavulanate	Amoxicillin–clavulanate
	Alternative:	Alternative:
	Piperacillin–tazobactam	Gentamicin + metronidazole
Appendicular-related sepsis [43]	Meropenem 7–10 days	Meropenem 5–7 days
	Alternative:	Alternative:
	Piperacillin–tazobactam +/− aminoglycoside	Piperacillin–tazobactam +/− aminoglycoside

* Up to 3 doses are recommended during surgery if the surgical intervention is extended. ^&^ Up to 14 days of total treatment if oral antibiotics are prescribed.

## Data Availability

The data presented in this study are available on request from the corresponding author. Raw data were generated as part of the routine work at Hospital Sant Joan de Déu and analysed therafter.

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
