# Peer review of "Effects of a Paediatric Antimicrobial Stewardship Program on Antimicrobial Use and Quality of Prescriptions in Patients with Appendix-Related Intraabdominal Infections"

_antibiotics, 2020, doi:10.3390/antibiotics10010005_

Round 1

Reviewer 1 Report

Thank you for your submission.

Antimicrobial stewardship is an important intervention to reduce the development of multidrug resistant microbes in patients admitted with potentially life-threatening infections.

The authors describe their comprehensive development of a pediatric antimicrobial stewardship program for appendix related intra-abdominal infections during a 72 month time-frame along with data in regard to various outcome aspects.  When comparing pre- and post-intervention outcome indicators they were not able to see differences in these outcomes at the end of the post intervention period, however the authors describe a trend for better outcomes.

Some additional specifics regarding the patient population and the surgical intervention may aid readers:

  1. How old were the patients in the different appendicitis classifications?
  2. Were different surgical interventions used open versus laparoscopic versus non-surgical?
  3. If cultures were done, which cultures (blood versus fluid etc)?
  4. Was appendicitis diagnosed using clinical exam or ultrasound or CT?
  5. Was there a change in vital signs based on age?

Non-surgical intervention for appendicitis using antibiotics has been shown to be non-inferior to surgical intervention for adults in the recently published CODA study [ The CODA Collaborative, A Randomized Trial Comparing Antibiotics with Appendectomy for Appendicitis, NEJM, Nov 12, 2020; 383, (20): 1907-1919.]. The time to source control for appendicitis is important and therefore you should look at the different interventions as stated under (2.) as part of your analysis. There may be also pediatric patients were treated only with antibiotics in your study. This should be evaluated and made clear to the readers.  Pediatric patients which received source control in different age groups with surgery also have different times to recovery following surgery and this may be important for the clinical decision to discontinue antibiotics as an important confounder in your data.

A clear description of the stewardship intervention should be placed in the earlier part of the manuscript along with a flow-chart of the changes the stewardship changes are. It appears in addition to the changes you show in table 2 you also implemented direct feedback, but it is not really how you did this. More details about these changes will be helpful. You also state that the surgeons were “satisfied” (Line 147. Did you have a survey? How was this measured?

Table 1. Please consider writing the indicator terms out. The abbreviations are a bit confusing.

Figure 1 is not really helpful and should be moved to the supplement.

Consider a different and additional analysis – Compare the 1st quarter of your pre intervention indicators and the last quarter of your post intervention period. This analysis should also be done comparing intervention compliance. As compliance changes it would be expected that outcomes may also improve. You show this in table S5, but you should add the changes in outcomes to this as well. Pre-intervention Compliance versus pre-intervention noncompliance outcomes information should be compared to the same at the post intervention time point. This should be summarized by the different relevant age groups. As you would look with this at COMPLIANCE versus NONCOMPLIANCE you would also not need to separate the different appendix related infection interventions separate. I would keep this information however as this shows how hard your team worked on this.

I think you have the necessary data to make these changes which will be helpful for the readers and also consider the implementation of similar interventions at their institutions.

Author Response

Reviewer 1

Thank you for your submission.

Antimicrobial stewardship is an important intervention to reduce the development of multidrug resistant microbes in patients admitted with potentially life-threatening infections.

The authors describe their comprehensive development of a pediatric antimicrobial stewardship program for appendix related intra-abdominal infections during a 72 month time-frame along with data in regard to various outcome aspects.  When comparing pre- and post-intervention outcome indicators they were not able to see differences in these outcomes at the end of the post intervention period, however the authors describe a trend for better outcomes.

We thank the reviewer for his/her very thoughtful and interesting inputs. We have done our best to address them all and we are sure that our article reads now better and will be of greater interest for the readers. Thanks !!!

Some additional specifics regarding the patient population and the surgical intervention may aid readers:

  1. How old were the patients in the different appendicitis classifications? Median (IQR) ages at diagnosis in the pre-intervention and in the post-intervention periods have been described in the Results section (line 94). Also, the median ages of the patients according to the different diagnostic groups have been added and compared between the 2 periods in new Table S1. Of note, no significant differences were observed in any of the analyses.

  1. Were different surgical interventions used open versus laparoscopic versus non-surgical?
  2. If cultures were done, which cultures (blood versus fluid etc)?
  3. Was appendicitis diagnosed using clinical exam or ultrasound or CT?

(answers to queries 2-4:) In our center, appendicitis is diagnosed on the basis of consistent history and physical examination, compatible selected laboratory studies and, in most cases, abdominal ultrasound evaluation.   

Blood cultures are usually obtained in case of fever (axillary ≥38 ºC), while peritoneal fluid cultures are only obtained when peritonitis is present.

The most common surgical approach is laparoscopy (93.0% and 93.3% in the pre-intervention and the post-intervention periods, respectively; p=0.067). The non-surgical approach in our center is very rare and usually reserved for older children with appendicular abscesses or masses that receive IV antibiotics initially and undergo 2nd-step surgery later on. The decision to treat conservatively is taken on an individual basis by the responsible surgeon.

Other than data about the surgical approach in each case, the rest of these data were not collected in our study.

We have therefore added a paragraph describing the diagnostic and therapeutic approach in our center in Methods (line 290), together with a new reference that validates this strategy (see below).

We have also added a sentence on the % of laparoscopy in the pre-intervention and post-intervention periods in the Results section (line 101).

We are aware that we are not completely addressing reviewer´s queries and we are happy to revisit this if needed.

Bundy DG, Byerley JS, Liles EA, Perrin EM, Katznelson J, Rice HE. Does this child have appendicitis? JAMA. 2007 Jul 25;298(4):438-51.

  1. Was there a change in vital signs based on age? Thanks for the suggestion. Unfortunately, these data are not available.

Non-surgical intervention for appendicitis using antibiotics has been shown to be non-inferior to surgical intervention for adults in the recently published CODA study [ The CODA Collaborative, A Randomized Trial Comparing Antibiotics with Appendectomy for Appendicitis, NEJM, Nov 12, 2020; 383, (20): 1907-1919.]. Thanks again for raising this very interesting issue. As previously commented, hardly any of the patients in our study was treated conservatively; this has been explained in the Results. Therefore, we were not able to assess the impact of this approach on antimicrobial use, length of stay or any of the other outcomes. Nevertheless, we think this issue merits further analysis and we´ve included a new paragraph in the Discussion and 2 new references (line 232).          

Huang L, Yin Y, Yang L, Wang C, Li Y, Zhou Z. Comparison of Antibiotic Therapy and Appendectomy for Acute Uncomplicated Appendicitis in Children: A Meta-analysis. JAMA Pediatr. 2017 May 1;171(5):426-434.

CODA Collaborative, Flum DR, Davidson GH, Monsell SE, Shapiro NI, Odom SR, Sanchez SE, Drake FT, Fischkoff K, Johnson J, et al. A Randomized Trial Comparing Antibiotics with Appendectomy for Appendicitis. N Engl J Med. 2020 Nov 12;383(20):1907-1919. 

The time to source control for appendicitis is important and therefore you should look at the different interventions as stated under (2.) as part of your analysis. There may be also pediatric patients were treated only with antibiotics in your study. This should be evaluated and made clear to the readers.  Pediatric patients which received source control in different age groups with surgery also have different times to recovery following surgery and this may be important for the clinical decision to discontinue antibiotics as an important confounder in your data. Thanks for raising this interesting point. As previously stated, the vast majority of patients in our study underwent laparoscopic appendectomy, without differences in the terapheutic approach depending on patients age or diagnosis. Also, the time between the initial consultation and surgery (which equals source control) was not collected. The lack of data and the homogeneous therapeutic approach in the vast majority of patients prevent us from performing this analysis. In future studies, we plan to include the time to source control as a relevant clinical variable for the outcomes of stewardship activities.    

A clear description of the stewardship intervention should be placed in the earlier part of the manuscript along with a flow-chart of the changes the stewardship changes are. It appears in addition to the changes you show in table 2 you also implemented direct feedback, but it is not really how you did this. More details about these changes will be helpful. Thanks for stressing this; to us, it is critical that the details of the intervention are clear to the readers. Due to Journal requirements, that require the Methods to be placed at the end of the manuscript, we cannot move these forward in the article. Nevertheless, we have done our best to explain the Methodology our stewardship program was based on. The changes are:

  • We have reordered some of the paragraphs to make clear that the update of the local guidelines was also part of the stewardship program (current line 320).
  • We have emphasized that daily feedback was given to the surgical team, either face to face or by telephone, besides the comments in the electronic chart (line 342).
  • We have created a flow-chart summarising the changes and the new interventions upon the stewardship program implementation (new Figure 2, line 358).

We honestly hope this now reads better.

You also state that the surgeons were “satisfied” (Line 147. Did you have a survey? How was this measured?) The reviewer is very right that this sentences do not belong to the Results section, as we did not objectively measure surgeons satisfaction, and this was rather a subjective perception of the ASP team. We have removed these sections from the Results, and have commented this in the limitations (line 268).

Table 1. Please consider writing the indicator terms out. The abbreviations are a bit confusing. Thanks for pointing this out. This has been ammended accordingly in Table 1.

Figure 1 is not really helpful and should be moved to the supplement.

We agree Figure 1 is a bit messy and does not add much to the Results. We have moved Figure 1 to the Online section (this is now Figure S4.) and have renamed previous Figure 2 accordingly.

Consider a different and additional analysis – Compare the 1st quarter of your pre intervention indicators and the last quarter of your post intervention period. This analysis should also be done comparing intervention compliance. As compliance changes it would be expected that outcomes may also improve. You show this in table S5, but you should add the changes in outcomes to this as well. Pre-intervention Compliance versus pre-intervention noncompliance outcomes information should be compared to the same at the post intervention time point. This should be summarized by the different relevant age groups. As you would look with this at COMPLIANCE versus NONCOMPLIANCE you would also not need to separate the different appendix related infection interventions separate. I would keep this information however as this shows how hard your team worked on this.

We thank the reviewer for this very interesting suggestion. Unfortunately, data about the compliance with the ASP recommendations or data on quality of prescriptions are not available for the pre-intervention period. Therefore, we cannot do the suggested analysis in this study. We shall take into consideration this suggestion for future studies comparing the efficacy of the ASP over time.

I think you have the necessary data to make these changes which will be helpful for the readers and also consider the implementation of similar interventions at their institutions.

Reviewer 2 Report

I have reviewed this manuscript describing a single center antimicrobial stewardship intervention to improve antibiotic prescribing in children with appendicitis. The before/after quasi-experimental design is well described. The use of interrupted time series analysis is appropriate for this design. I have only few comments regarding this manuscript.

  • It would be informative to list the length of therapy and length of stay before and after the intervention for both phlegmonous and complicated appendicitis in a table.
  • I would add to the Discussion section the potential long-term benefits of reducing piperacillin/tazobactam (an antipseudomonal agent) use in this population or in the hospital in general such as potential improvement in Pseudomonas aeruginosa resistance rates or reduction in C. difficile infection as demonstrated in prior studies.
  • It seems essential to highlight the multidisciplinary collaboration in this antimicrobial stewardship intervention with the surgeons and other specialists.
  • It would be useful to briefly justify the rational for the selected antimicrobial regimens in the antimicrobial stewardship intervention. For example, do the authors know E. coli susceptibility rates to cefoxitin and ceftriaxone in this or overall hospital population? Why was meropenem recommended for sepsis? Was that based on high prevalence of ESBLs?
  • How was sepsis defined in “appendicular-related sepsis”?
  • Section 4.4 should be labelled “definitions” not “outcomes”.
  • Correct spelling error in Table 2 (4th row in table).

Author Response

Reviewer 2

I have reviewed this manuscript describing a single center antimicrobial stewardship intervention to improve antibiotic prescribing in children with appendicitis. The before/after quasi-experimental design is well described. The use of interrupted time series analysis is appropriate for this design. I have only few comments regarding this manuscript.

We thank the reviewer for his/her comments, which we have tried to address in the new version of the manuscript. We hope this now reads better and is of greater interest for the readers. Thanks again.

  • It would be informative to list the length of therapy and length of stay before and after the intervention for both phlegmonous and complicated appendicitis in a table. Thanks for the suggestion. These data were available in Supplementary Table 3 (antimicrobial use in length of therapy, in days, according to the different diagnoses) and Supplementary Table 4 (length of stay, in days, according to the different diagnoses). We hope these are the data the reviewer asks for, but are happy to review it again in case we misunderstood.
  • I would add to the Discussion section the potential long-term benefits of reducing piperacillin/tazobactam (an antipseudomonal agent) use in this population or in the hospital in general such as potential improvement in Pseudomonas aeruginosa resistance rates or reduction in C. difficile infection as demonstrated in prior studies. Thanks for the suggestion. This had already been mentioned in the Discussion, but we agree that it is worth emphasizing. We have added a sentence describing the decrease in these 2 pathogens infection rates (line 218) and have added as well 2 new references.
  • It seems essential to highlight the multidisciplinary collaboration in this antimicrobial stewardship intervention with the surgeons and other specialists. We cannot agree more with the reviewer. This has been emphasized in the Discussion (line 259).  
  • It would be useful to briefly justify the rational for the selected antimicrobial regimens in the antimicrobial stewardship intervention. For example, do the authors know E. coli susceptibility rates to cefoxitin and ceftriaxone in this or overall hospital population? Why was meropenem recommended for sepsis? Was that based on high prevalence of ESBLs? Thanks for raising this point. The local guidelines that were updated on January 2017 were based on the Guidelines by the Surgical Infection Society and the Infectious Diseases Society of America (ref. 16 in the article). This has been clarified in the text (line 324).
  • How was sepsis defined in “appendicular-related sepsis”? Appendicular-related sepsis was defined according to sepsis criteria defined elsewhere (see reference below) of appendicular origin. The new reference has been added in Table 2.

Goldstein B, Giroir B, Randolph A. International Consensus Conference on Pediatric S. International pediatric sepsis consensus conference: definitions for sepsis and organ dysfunction in pediatrics. Pediatr Crit Care Med. (2005) 6:2–8.

  • Section 4.4 should be labelled “definitions” not “outcomes”. Thanks for spotting this error, which has been ammended (line 362).
  • Correct spelling error in Table 2 (4th row in table). Thanks for spotting this error, which has been ammended (Table 2).